

# Nomogram model for predicting the risk of post-stroke depression based on clinical characteristics and DNA methylation

Shihang Luo[1], Fan Liu[2], Qiao Liao[2], Hengshu Chen[2], Tongtong Zhang[3,4] and Rui Mao[2]

[1] The Affiliated Nanhua Hospital, Department of Neurology, Hengyang Medical School, University of South China, Hengyang, Hunan, China

[2] Xiangya Hospital, Central South University, Changsha, Hunan Province, China

[3] Medical Research Center, The Third People's Hospital of Chengdu, Affiliated Hospital of Southwest Jiaotong University & The Second Affiliated Hospital of Chengdu, Chongqing Medical University, Chengdu, China

[4] Center of Gastrointestinal and Minimally Invasive Surgery, Department of General Surgery, The Third People's Hospital of Chengdu, Affiliated Hospital of Southwest Jiaotong University & The Second Affiliated Hospital of Chengdu, Chongqing Medical University, Chengdu, China

Corresponding authors
Tongtong Zhang,
mr1995@my.swjtu.edu.cn
Rui Mao, 218102100@csu.edu.cn

## ABSTRACT

**Objective**. To construct a comprehensive nomogram model for predicting the risk of post-stroke depression (PSD) by using clinical data that are easily collected in the early stages, and the level of DNA methylation, so as to help doctors and patients prevent the occurrence of PSD as soon as possible.

**Methods**. We continuously recruited 226 patients with a history of acute ischemic stroke and followed up for three months. Socio-demographic indicators, vascular-risk factors, and clinical data were collected at admission, and the outcome of depression was evaluated at the third month after stroke. At the same time, a DNA-methylation-related sequencing test was performed on the fasting peripheral blood of the hospitalized patients which was taken the morning after admission.

**Results**. A total of 206 samples were randomly divided into training dataset and validation set according to the ratio of 7:3. We screened 24 potentially-predictive factors by Univariate logistic regression and least absolute shrinkage and selection operator (LASSO) regression analysis, and 10 of the factors were found to have predictive ability in the training set. The PSD nomogram model was established based on seven significant variables in multivariate logistic regression. The consistency statistic (C-index) was as high as 0.937, and the area under curve (AUC) in the ROC analysis was 0.933. Replication analysis results in the validation set suggest the C-index was 0.953 and AUC was 0.926. This shows that the model has excellent calibration and differentiating abilities.

**Conclusion**. Gender, Rankin score, history of hyperlipidemia, time from onset to hospitalization, location of stroke, National Institutes of Health Stroke scale (NIHSS) score, and the methylation level of the cg02550950 site are all related to the occurrence of PSD. Using this information, we developed a prediction model based on methylation characteristics.

## INTRODUCTION

Cerebrovascular accidents, more commonly referred to as strokes, represent the second leading global cause of mortality and serve as a predominant etiological agent of enduring disability among middle-aged (*Campbell et al., 2019*) and geriatric populations. Among the multifaceted complications arising from strokes, post-stroke depression (PSD) has witnessed an escalating prevalence within clinical scenarios (*Lenzi, Altieri & Maestrini, 2008*). The intricate etiology of PSD encompasses socio-psychological dynamics, pathophysiological alterations, and myriad other contributory factors. This complexity underpins the prevailing absence of a coherent, efficacious clinical approach towards PSD, often culminating in suboptimal therapeutic outcomes for the afflicted (*Starkstein, Mizrahi & Power, 2008*; *Trusova & Levin, 2019*).

Concomitantly, PSD stands as a formidable impediment to the optimal recovery of neurocognitive function subsequent to a stroke. Such impediments not only attenuate the recuperation from cerebral-neurological deficits but might exacerbate post-stroke symptomatology, profoundly undermining day-to-day functional capacity and vocational engagement of stroke survivors. Consequent to these ramifications, pertinent research elucidates that the mortality indices among PSD sufferers significantly surpass those of their non-depressed counterparts post-stroke. Specifically, within a decade following the ischemic incident, stroke survivors exhibiting depressive symptoms faced a mortality rate amplifying by over fivefold in comparison to their non-depressed counterparts (*Levada & Slivko, 2006*).

Given the often-late onset of PSD—frequently materializing a month or longer post-stroke—and its potential under-recognition in clinical settings, there appears to be a diminished emphasis on its clinical significance within numerous medical establishments. When juxtaposed against non-complicated stroke cases, PSD typically portends elevated mortality, compromised neurological recuperation, pronounced cognitive deficits, and diminished life quality. In summation, the profound clinical implications of PSD underscore the imperative for healthcare practitioners to be adeptly acquainted with its risk factors, facilitating the proactive initiation of preventative and therapeutic modalities for susceptible individuals (*Arseniou, Arvaniti & Samakouri, 2011*).

DNA methylation is an epigenetic modification that occurs at cytosine-phosphate-guanine (CpG) sites. This modification has been shown to play an important role in regulating gene expression, RNA processing, and protein function. Studies on DNA methylation have shown rich and complex prospects for epigenetic gene regulation in the central nervous system. In view of the risk that DNA methylation may lead to mental illness, manipulating methylation levels is a promising method in the development of new treatments (*Guidotti & Grayson, 2014*). In recent years, with the development of high-throughput-sequencing technology, the number of bioinformatics-related clinical studies using the data generated with this technology has increased rapidly, and with these techniques it is possible to measure the epigenetic modification of the whole genome. Previous studies have shown that the level of DNA methylation at a specific CpG site in the promoter region of the brain-derived neurotrophic factor (BDNF) gene is related
to the occurrence and treatment of PSD in mice, and can be regulated by fluoxetine (*Jin et al., 2017*). But so far, the relationship between PSD and DNA methylation is not supported by a large quantity of research evidence, so we have comprehensively studied the methylation and expression profiles of PSD-related genes, and evaluated their predictive value. A model was developed to predict the occurrence of PSD using the combination of methylation sites and clinical characteristics of the patients. Our findings indicate that this specific methylation site has considerable potential for PSD prediction and provides strong evidence for our ability to find therapeutic targets for PSD.

In order to study the potential DNA-methylation sites related to the occurrence of PSD, this study established an early-warning screening model and an outcome-prediction model around the pathogenesis of PSD epigenetics, and other relevant clinical indicators. By screening the high-risk population that matched the model, most of the patients with potential PSD could be identified, making it possible to improve the early-recognition rate of PSD, and to use corresponding early prevention and intervention measures to promote the overall rehabilitation of patients (*Burton & Gibbon, 2005*; *Allida et al., 2020*).

## PATIENTS AND METHODS

### Patients

Patients with acute stroke were admitted to the Department of Neurology, Xiangya Hospital of Central South University from June 2019 to May 2021. About 10 ml of fasting peripheral venous blood was taken the morning after admission and stored in a disposable, vacuum blood collection and coagulation-promoting tube, and then the serum was separated by centrifuge (3,000 rpm, 10 min) within 2 h or temporarily stored in 4 °C refrigerator and centrifuged within 4 h; after centrifugation the supernatant was discarded and stored in the freezer at $-80$ °C. At the same time, the clinical data for the hospitalized patients were collected. Depressive symptoms were diagnosed with the fifth edition of the Diagnostic and Statistical Manual of Mental Disorders (DSM-5) one month ($\pm3$ days) after stroke. This study was approved by the Ethics Committee of Xiangya Hospital of Central South University (ethics approval number: 201910842). All participants signed informed consent forms.

The inclusion criteria were as follows: (1) age >18 years old; (2) the patient met the diagnostic criteria of stroke from the Fourth Chinese Academic Conference on Cerebrovascular Diseases; had responsible lesions on computerized tomography (CT) or magnetic resonance imaging (MRI) scans; accompanied by sudden general or focal neurological impairment, lasting more than twenty-four hours; (3) included cerebral infarction, cerebral hemorrhage, and cerebral infarction with remission after thrombolysis; (4) a time from stroke onset to hospitalization of no more than fourteen days; (5) informed consent and permission to keep the relevant blood samples for experimental purposes.

The exclusion criteria were: (1) those who had a history of depression, dementia, and other mental illness, or who had taken related antidepressants and other antipsychotic drugs and instruments; (2) those who were unable to complete the follow-up due to hearing or expression disorders, disturbance of consciousness, or a mini-mental state examination

(MMSE) score of <17; (3) TIA and subarachnoid hemorrhage; (4) other nervous system diseases, such as epilepsy; (5) neurological impairment due to other causes, such as a brain tumor; (6) other major diseases (such as cancer), or death occurring before the follow up.

## Collection of clinical features

On the second day after admission, the basic and clinical data of the patients were collected, including age, sex, education, occupation, work, working method, marital status, number of children, smoking history, drinking history, diabetes, hypertension, hyperlipidemia, operation history, interests, time from onset to hospitalization, stroke location, and other clinical indicators. Then the National Institutes of Health Stroke scale (NIHSS) was used to evaluate the corresponding neurological impairment of stroke patients, the Barthel Index (BI) rating and modified Rankin scales were used to evaluate the patients' ability of living, and the mini-mental state examination (MMSE) was used to determine whether the patients had cognitive impairment.

## DNA methylation sequencing

We randomly selected 10 PSD patients to use as the case group, including five males and five females, and 10 non-PSD patients were selected for the control group, who were matched according to age and sex. We used the Infinium MethylationEPIC BeadChip (a DNA methylation 850K chip, produced by Illumina). Of the original 450K chip sites, 91% were included in the data, to make full use of the original 450K data, and an additional 350K sites in the enhancer region were added, which can be used for quantitative methylation detection of a single CpG site in normal samples. The chip's comprehensive genome coverage is high, and the use of two kinds of probes can maximize the detection range at the same time. The exact site of the methylation can be detected directly, which is very useful for the initial screening of this experiment.

After conducting a rigorous screening of differential methylation sites utilizing the 850K methylation chip, the disparities in methylation between two defined groups were assessed through the computation of the mean difference (calculation method: mean of the PSD group samples minus the mean of the non-PSD group samples). Employing statistical significance based on the $P$-values, the site with the most pronounced difference in methylation degree, along with three sites exhibiting lower $P$-values, were isolated. According to the technical prerequisites of MethylTarget second-generation sequencing, an extension of 30 kb to either side of each methylation site is necessitated. By leveraging MethylTarget technology, facilitated by the second-generation sequencing platform, several distinct CpG islands can be concurrently captured and sequenced. Furthermore, the methylation degree of each individual CpG site can be accurately ascertained utilizing high-depth sequencing data.

## Statistical analysis
### Screen predictors

Least absolute shrinkage and selection operator (LASSO) regression analysis was carried out by R software (version 3.6.1; *R Core Team, 2019*). The risk factors that needed to be analyzed were cross-verified, and the lambda value with the smallest error in the cross validation

was obtained. The LASSO regression method was used to construct the penalty function to obtain a better model. While compressing the absolute value of the sample regression coefficient, regression coefficients with very small absolute values are directly penalized to 0, thus the prediction factors that have great influence on the results are selected, and the factors with small influence are excluded. This method retains the advantage of subset contraction, and is a biased estimation for dealing with complex collinear data (*Belhechmi et al., 2020*). However, the LASSO regression coefficient map of predictive factors mainly depends on lambda, and the cross-validation method is used to draw the vertical line at loglambda, in which the coefficients of the best lambda-generated predictive factors are all non-zero. These screened factors are all possible to model.

### Establish the prediction model of the nomogram

The independent risk factors of PSD were screened out using multi-factor logistic regression analysis ($P < 0.05$). According to the final results of the multi-factor analysis, the odds ratio (OR) of each risk factor to PSD is calculated and expressed with a 95% confidence interval, and the risk prediction nomogram model of PSD is established by R software. In the nomogram model, the probability of PSD risk can be obtained by calculating the sum of scores corresponding to each predictor.

### Validate the PSD risk-prediction model

In this study, a variety of methods are used to verify the accuracy and differentiation of the model. Calibration is evaluated by comparing the predicted probability derived from the nomogram with the actual probability curve, using internal bootstrap verification; that is, the size of the random sample extracted by the replacement method from the original data set is the same as that of the original queue. However, in this new sample, patient L may appear five times, while patient M may appear ten times. Although each patient has the same sampling probability, random chance can lead to such unbalanced results. In fact, each bootstrap sample usually contained at least one raw observation data of about 2 beat 3. In this study, this process was repeated 1,000 times to verify that the average performance index of the new queue model was comparable to the performance of the model established in the study, and a $P$-value and confidence interval were used to evaluate the accuracy of the model. Then, the receiver operating characteristic (ROC) curve was obtained and the value of the area under the curve (AUC) was calculated. When the value of the AUC is greater than 0.5, the model has a degree of discrimination;the higher the value of AUC, the better the discrimination of the model.

### The drawing of the decision curve

Finally, the clinical prediction practicability of the model was evaluated by using the decision curve; and the clinical validity of the nomogram established by the research results was determined by quantifying the net income under different threshold probabilities in the queue and analyzing the resulting decision curves.

**Table 1  Second-generation sequencing verification input fragment.**

| Fragment name | Genes | Chromosome number | Starting position | Termination position |
|---|---|---|---|---|
| cg02550950-17 | HECW2 | 2 | 196257312 | 196257552 |
| cg03329597-19 | MYH15 | 3 | 108406600 | 108406749 |
| cg13557709-20 | GTF3A | 13 | 27425254 | 27425464 |
| cg25290307-18 | EMID2 | 7 | 101362483 | 101362709 |

# RESULTS

Consequently, a total of four significant, differential methylation sites were identified: cg03329597, cg13557709, cg02550950, and cg25290307, each localized on the MYH15, GTF3A, HECW2, and EMID2 genes, respectively. The explicit fragment information is delineated in Table 1. In this particular experiment, samples from 226 patients, encompassing both experimental and control groups, were meticulously collected to validate and discern the differential methylation sites.

## Analysis of the data of patients enrolled in the group

A total of 226 patients with acute stroke were enrolled in this study, aged from 33 to 77 years old, including 112 cases in the PSD group and 114 cases in the non-PSD group. A total of 24 predictive factors were included. The general data and grading criteria of patients are shown in Table 2. 206 samples were randomly divided into training dataset ($n = 144$) and validation set ($n = 62$) according to the ratio of 7:3. Univariate logistic regression analysis on training dataset prompts that gender, education level, time from onset to hospitalization, history of hyperlipidemia, nature of the work, location of stroke, NIHSS score, BI score, Rankin score, and *HECW2* gene methylation were all statistically significant ($P < 0.05$), as shown in Table 3.

## Establishment and validation of the PSD prediction model
### Screening predictors with LASSO regression analysis

We used the occurrence of depression as the outcome variable, and the 10 factors with $P < 0.05$ in the univariate analysis as independent variables. Through the analysis of the LASSO regression model (lambda.min pattern), all ten predictors (non-zero regression coefficients) with modeling ability were selected, as shown in Fig. 1. Figure 1A shows the LASSO-regression coefficient map corresponding to 10 factors—where each curve corresponds to a factor—in which the ordinate is the regression coefficient of the predictor and the abscissa is log lambda. Figure 1B shows the binomial deviation curve; the lowest point of this curve is the optimal parameter, lambda.

### Establishment of a nomogram model for predicting the risk of PSD

Ten predictors screened by LASSO regression were used as independent variables for multivariate-logistic regression analysis in the training set. The results showed that seven of the variables were independent impact factors for depression in stroke patients, as shown in Table 4. Then seven independent impact factors were used to establish a nomogram

**Table 2  Differences in predictive factors between the PSD group and the non-PSD group.**

| Prediction factors | Variable assignment | PSD | N-PSD | Total |
|---|---|---|---|---|
| **Gender** | | | | |
| Male | 0 | 69 | 90 | 159 |
| Female | 1 | 43 | 24 | 67 |
| **Age** | | | | |
| ≥60 | 0 | 49 | 46 | 95 |
| 45–60 | 1 | 49 | 49 | 98 |
| <45 | 2 | 14 | 19 | 33 |
| **Cultural level** | | | | |
| Primary or below | 0 | 41 | 21 | 62 |
| Junior high school | 1 | 54 | 63 | 117 |
| College or above | 2 | 17 | 30 | 47 |
| **Occupation** | | | | |
| Farmer | 0 | 34 | 30 | 64 |
| Civil servant or staff | 1 | 33 | 46 | 79 |
| Service industry | 2 | 21 | 19 | 40 |
| Others | 3 | 24 | 19 | 43 |
| **Working condition** | | | | |
| Full time | 0 | 80 | 79 | 159 |
| retire | 1 | 16 | 27 | 43 |
| Part-time or other | 2 | 16 | 8 | 24 |
| **The mode of work** | | | | |
| Physical labor | 0 | 64 | 51 | 115 |
| Mental labor | 1 | 24 | 45 | 69 |
| Physical is equal to mental labor | 2 | 24 | 18 | 42 |
| **Marital status** | | | | |
| Married | 0 | 104 | 105 | 209 |
| Unmarried | 1 | 2 | 3 | 5 |
| Divorced or widowed | 2 | 6 | 6 | 12 |
| **Number of children** | | | | |
| 0 | 0 | 2 | 3 | 5 |
| 1 | 1 | 33 | 31 | 64 |
| 2 or more | 2 | 77 | 80 | 157 |
| **Life style** | | | | |
| Live alone | 0 | 15 | 9 | 24 |
| With a Spouse | 1 | 33 | 48 | 81 |
| Spouse and children | 2 | 55 | 50 | 105 |
| Others | 3 | 9 | 7 | 16 |
| **Smoking history** | | | | |
| Quit smoking | 0 | 9 | 10 | 19 |
| Less than 20 cigarettes per day | 1 | 22 | 37 | 59 |

| Prediction factors | Variable assignment | PSD | N-PSD | Total |
|---|---|---|---|---|
| More than 20 cigarettes per day | 2 | 23 | 28 | 51 |
| No smoking | 3 | 58 | 39 | 97 |
| **Drinking history** | | | | |
| Have given up drinking | 0 | 13 | 12 | 25 |
| A small amount | 1 | 24 | 37 | 61 |
| Drinking more than 50g per day | 2 | 15 | 22 | 37 |
| No drinking | 3 | 60 | 43 | 103 |
| **Diabetes history** | | | | |
| None | 0 | 70 | 74 | 144 |
| With poor or untreated treatment | 1 | 28 | 26 | 54 |
| With good treatment | 2 | 11 | 12 | 23 |
| Do not know the history | 3 | 3 | 2 | 5 |
| **History of hypertension** | | | | |
| None | 0 | 30 | 35 | 65 |
| With poor or untreated treatment | 1 | 49 | 34 | 83 |
| With good treatment | 2 | 24 | 37 | 61 |
| Do not know the history | 3 | 9 | 8 | 17 |
| **History of hyperlipidemia** | | | | |
| None | 0 | 43 | 65 | 108 |
| With poor or untreated treatment | 1 | 38 | 25 | 63 |
| With good treatment | 2 | 6 | 13 | 19 |
| Do not know the history | 3 | 25 | 11 | 36 |
| **History of interest** | | | | |
| None | 0 | 63 | 67 | 130 |
| Yes | 1 | 49 | 47 | 96 |
| **The time from onset to hospitalization** | | | | |
| Less than or equal to 3 days | 0 | 33 | 50 | 83 |
| 4–7 days | 1 | 43 | 45 | 88 |
| 8–11 days | 2 | 36 | 19 | 55 |
| **Stroke area** | | | | |
| Anterior circulation | 0 | 88 | 53 | 141 |
| Posterior circulation | 1 | 22 | 54 | 76 |
| Both circulation are involved | 2 | 2 | 7 | 9 |
| **NIHSS score** | | | | |
| 0–1 | 0 | 9 | 51 | 60 |
| 2–4 | 1 | 39 | 43 | 82 |
| 5–15 | 2 | 64 | 20 | 84 |
| **Bi Index** | | | | |
| 100 Points | 0 | 38 | 61 | 99 |
| 61–99 Points | 1 | 33 | 29 | 62 |
| 41–60 Points | 2 | 4 | 7 | 11 |
| ≤40 Points | 3 | 37 | 17 | 54 |
| **Rankin** | | | | |
| 0–1 Points | 0 | 34 | 60 | 94 |

**Table 2** (*continued*)

| Prediction factors | Variable assignment | PSD | N-PSD | Total |
|---|---|---|---|---|
| 2–3 Points | 1 | 35 | 32 | 67 |
| ≥4 Points | 2 | 43 | 22 | 65 |
| **cg02550950** | | | | |
| Hypomethylation | 0 | 1 | 28 | 29 |
| Hypermethylation | 1 | 74 | 68 | 142 |
| More than Hypermethylation | 2 | 37 | 18 | 55 |
| **cg03329597** | | | | |
| Hypomethylation | 0 | 53 | 63 | 116 |
| Hypermethylation | 1 | 47 | 40 | 87 |
| More than Hypermethylation | 2 | 12 | 11 | 23 |
| **cg13557709** | | | | |
| Hypomethylation | 0 | 28 | 27 | 55 |
| Hypermethylation | 1 | 84 | 87 | 171 |
| More than Hypermethylation | 2 | 0 | 0 | 0 |
| **cg25290307** | | | | |
| Hypomethylation | 0 | 25 | 37 | 62 |
| Hypermethylation | 1 | 87 | 77 | 164 |
| More than Hypermethylation | 2 | 0 | 0 | 0 |

**Notes.**

Hypomethylation (methylation rate <0.334); Hypermethylation methylation rate: 0.334–0.666; More than Hypermethylation (methylation rate ≥0.667).

model to predict the risk of PSD in 3 months, as shown in Fig. 1C. The C-index of the nomogram model accrodding to the Hmisc C-index analysis was 0.908 (95% CI [0.901–0.915]). According to the proposed nomogram, we can estimate the PSD rate in patients. For example, a male patient (patient id 53, corresponds to 306 points) with 9 days from onset to hospitalization (corresponds to 56 points), 5 Rankin point (corresponds to 39 points), both posterior and anterior circulation were involved (corresponds to 13 points), do not know the history of hyperlipidemia (corresponds to 36 points), 8 NIHSS score (corresponds to 67 points), and Hypermethylation of cg02550950 (corresponds to 95 points). The calculation according to the proposed nomogram is thus 237 points, predicting a PSD rate of 92.7%. Consistent with the predictions, she did develop PSD.

*Validation of the PSD occurrence risk-prediction model*

The calibration curve is used to evaluate and predict the risk-nomogram model of PSD occurrence, in which the $x$-axis is the predicted likelihood of PSD, and the $y$-axis is the likelihood of receiving a PSD diagnosis. The diagonal, dotted line represents the ideal model for prediction, and the solid line is the nomogram model obtained in this study. The more the solid line and the dashed line fit, the better the prediction effect. Figure 1D shows the accuracy of the stroke-patient, depression-risk nomogram model in this cohort. In this study, the Bootstrap method was used for internal bootstrap verification, and the C-index of the nomogram model wass 0.859. The results indicate that the model has a good degree of differentiation. Finally, the ROC curve was drawn to evaluate the statistical calculation

**Table 3** Univariate regression analysis of the PSD group and the non-PSD group.

| | Odds ratio (95%CI) | P-value | Significance |
|---|---|---|---|
| **Age** | | | |
| ≥60 | ref | ref | |
| 45–60 | 1.37(0.67–2.82) | 3.88E−01 | ns |
| <45 | 0.83(0.31–2.24) | 7.14E−01 | ns |
| **Bi Index** | | | |
| 100 Points | ref | ref | |
| 61–99 Points | 1.57(0.68–3.58) | 2.89E−01 | ns |
| 41–60 Points | 1.57(0.38–6.4) | 5.33E−01 | ns |
| ≤40 Points | 6.71(2.52–17.88) | 1.42E−04 | * |
| **cg02550950** | | | |
| Hypomethylation | ref | ref | |
| Hypermethylation | 25.4(3.28–196.52) | 1.95E−03 | * |
| More than Hypermethylation | 45.23(5.48–373.45) | 4.01E−04 | * |
| **cg03329597** | | | |
| Hypomethylation | ref | ref | |
| Hypermethylation | 1.4(0.69–2.83) | 3.52E−01 | ns |
| More than Hypermethylation | 0.97(0.31–3.06) | 9.59E−01 | ns |
| **cg13557709** | | | |
| Hypomethylation | ref | ref | |
| Hypermethylation | 1.14(0.52–2.46) | 7.47E−01 | ns |
| **cg25290307** | | | |
| Hypomethylation | ref | ref | |
| Hypermethylation | 2.12(0.98–4.56) | 5.55E−02 | ns |
| **Cultural level** | | | |
| Primary or below | ref | ref | |
| Junior high school | 0.46(0.21–1.04) | 6.26E−02 | ns |
| College or above | 0.19(0.07–0.54) | 1.84E−03 | * |
| **Drinking history** | | | |
| Have given up drinking | ref | ref | |
| A small amount | 0.56(0.17–1.82) | 3.35E−01 | ns |
| Drinking more than 50g per day | 0.67(0.19–2.36) | 5.30E−01 | ns |
| No drinking | 1.21(0.4–3.61) | 7.37E−01 | ns |
| **Gender** | | | |
| Male | ref | ref | |
| Female | 2.43(1.18–5.01) | 1.59E−02 | * |
| **Diabetes history** | | | |
| None | ref | ref | |
| With poor or untreated treatment | 1.9(0.86–4.24) | 1.14E−01 | ns |
| With good treatment | 0.74(0.23–2.39) | 6.15E−01 | ns |
| Do not know the history | 2(0.32–12.55) | 4.60E−01 | ns |

**Table 3** (*continued*)

| | Odds ratio (95%CI) | *P*-value | Significance |
|---|---|---|---|
| **History of hyperlipidemia** | | | |
| None | ref | ref | |
| With poor or untreated treatment | 1.97(1.08–4.36) | 9.70E−03 | * |
| With good treatment | 0.51(0.13–2.04) | 3.38E−01 | ns |
| Do not know the history | 2.28(1.39–5.81) | 8.48E−03 | * |
| **History of hypertension** | | | |
| None | ref | ref | |
| With poor or untreated treatment | 2.31(0.98–5.31) | 8.33E−02 | ns |
| With good treatment | 0.48(0.18–1.23) | 1.24E−01 | ns |
| Do not know the history | 1.07(0.25–4.55) | 9.30E−01 | ns |
| **History of interest** | | | |
| None | ref | ref | |
| Yes | 0.86(0.44–1.67) | 6.59E−01 | ns |
| **Life style** | | | |
| Live alone | ref | ref | |
| With a Spouse | 0.39(0.12–1.27) | 1.17E−01 | ns |
| Spouse and children | 0.69(0.22–2.16) | 5.22E−01 | ns |
| others | 0.67(0.14–3.09) | 6.04E−01 | ns |
| **Stroke area** | | | |
| Anterior circulation | ref | ref | |
| Posterior circulation | 0.3(0.14–0.63) | 1.53E−03 | * |
| Both circulation are involved | 0.37(0.06–2.14) | 2.68E−01 | ns |
| **Marital status** | | | |
| Married | ref | ref | |
| Unmarried | 0(0-Inf) | 9.88E−01 | ns |
| Divorced or widowed | 1.75(0.47–6.47) | 4.05E−01 | ns |
| **The mode of work** | | | |
| Physical labor | ref | ref | |
| Mental labor | 0.32(0.14–0.71) | 5.16E−03 | * |
| Physical is equal to mental labor | 1.19(0.49–2.86) | 7.03E−01 | ns |
| **NIHSS score** | | | |
| 0–1 | ref | ref | |
| 2–4 | 4.8(1.63–14.15) | 4.45E−03 | * |
| 5–15 | 18.71(6.02–58.12) | 4.10E−07 | * |
| **Occupation** | | | |
| Farmer | ref | ref | |
| Civil servant or staff | 0.49(0.2–1.21) | 1.22E−01 | ns |
| Service industry | 0.69(0.26–1.83) | 4.60E−01 | ns |
| Others | 0.95(0.36–2.5) | 9.24E−01 | ns |
| **Rankin** | | | |
| 0–1 Points | ref | ref | |
| 2–3 Points | 2.04(0.91–4.57) | 8.22E−02 | ns |
| ≥4 Points | 5.94(2.44–14.46) | 8.66E−05 | * |

**Table 3** (*continued*)

| | Odds ratio (95%CI) | *P*-value | Significance |
|---|---|---|---|
| **Smoking history** | | | |
| Quit smoking | ref | ref | |
| Less than 20 cigarettes per day | 0.99(0.21–4.7) | 9.88E−01 | ns |
| More than 20 cigarettes per day | 1.23(0.25–6.02) | 8.00E−01 | ns |
| No smoking | 2.18(0.48–9.96) | 3.15E−01 | ns |
| **The time from onset to hospitalization** | | | |
| Less than or equal to 3 days | ref | ref | |
| 4–7 days | 1.13(0.51–2.5) | 7.58E−01 | ns |
| 8–11 days | 5.13(2.04–12.91) | 5.11E−04 | * |
| **Working condition** | | | |
| Full time | ref | ref | |
| Retire | 0.77(0.32–1.86) | 5.57E−01 | ns |
| Part-time or other | 2.28(0.84–6.2) | 1.07E−01 | ns |

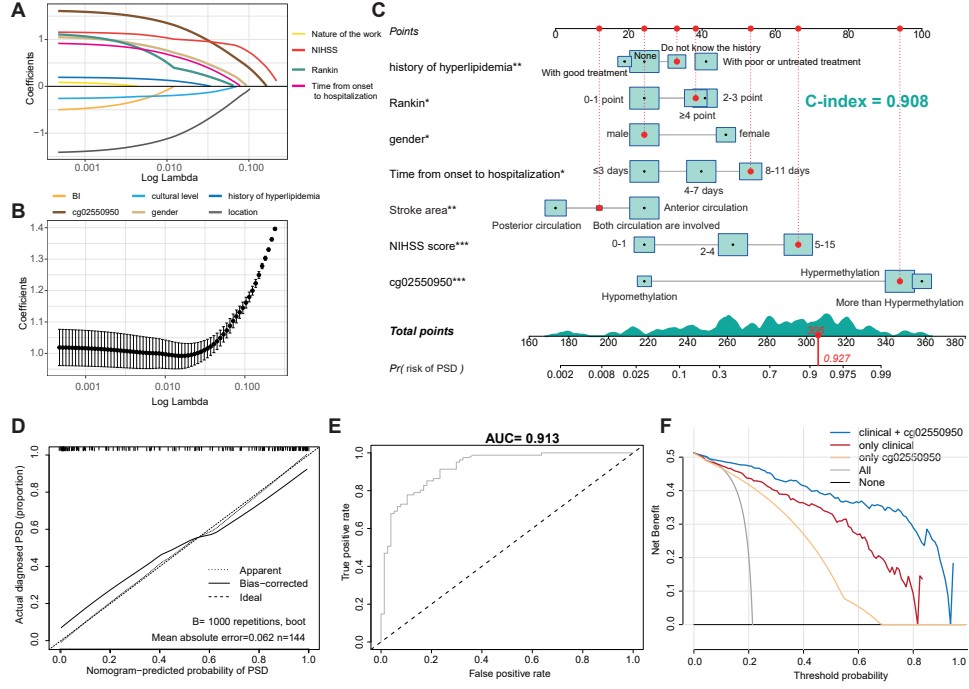

**Figure 1  Construction of a nomogram model for predicting the risk of PSD.** (A) LASSO regression curve of PSD. (B) LASSO regression coefficient diagram of PSD. (C) Nomogram Model for predicting the risk of PSD. (D) Calibration Diagram of PSD occurrence risk Diagram Model. (E) ROC Diagram of the Nomogram Model for predicting the occurrence risk of PSD. (F) Clinical decision curve of PSD occurrence risk prediction model.

**Table 4  Results of PSD multivariate logistic regression analysis.**

| Prediction factors | $\beta$ (95%CI) | P Value |
|---|---|---|
| **Intercept** | $-8.24(-11.91, -4.57)$ | <0.001[***] |
| **Gender** | | |
| Male | ref | |
| Female | 1.88(0.46, 3.29) | 0.009[*] |
| **History of hyperlipidemia** | | |
| None | ref | |
| With poor or untreated treatment | 1.02 (0.32, 3.21) | 0.018[**] |
| With good treatment | $-0.3(-1.86, 1.26)$ | 0.705 |
| Do not know the history | 0.68($-0.74$, 2.1) | 0.347 |
| **Cultural level** | | |
| Primary or below | ref | |
| Junior high school | $-0.56(-2.01, 0.88)$ | 0.446 |
| College or above | $-1.16(-3.16, 0.83)$ | 0.253 |
| **Time from onset to hospitalization** | | |
| 0–3 days | ref | |
| 4–7 days | 1.11($-0.01$, 2.23) | 0.052 |
| 8–11 days | 2.21(0.56, 3.87) | 0.009[*] |
| **NIHSS score** | | |
| 0–1 Points | ref | |
| 2–4 Points | 1.75(0.26, 3.24) | 0.021[*] |
| 5–15 Points | 3.62(1.69, 5.55) | <0.001[***] |
| **Stroke area** | | |
| Anterior circulation | ref | |
| Posterior circulation | $-1.8(-3, -0.61)$ | 0.003[**] |
| Both circulation are involved | $-1.32(-7.61, 4.97)$ | 0.681 |
| **cg025509501** | | |
| Hypomethylation | ref | |
| Hypermethylation | 5.45(2.71, 8.19) | <0.001[***] |
| More than Hypermethylation | 5.54(2.75, 8.33) | <0.001[***] |
| **Rankin** | | |
| 0–1 Points | ref | |
| 2–3 Points | 1.88(0.21, 3.54) | 0.027[*] |
| $\geq$4 Points | 2.36($-0.09$, 4.81) | 0.059 |
| **The mode of work** | | |
| Physical labor | ref | |
| Mental labor | 0.94($-0.56$, 2.45) | 0.220 |
| Physical is equal to mental labor | 0.37($-1.22$, 1.96) | 0.647 |
| **Bi Index** | | |
| 100 Points | ref | |

**Table 4** (*continued*)

| Prediction factors | β (95%CI) | P Value |
|---|---|---|
| 61–99 Points | −0.52(−2.11, 1.07) | 0.523 |
| 41–60 Points | −4.14(−7.42, 0.87) | 0.053 |
| ≤40 Points | −1.7(−4.22, 0.82) | 0.185 |

**Notes.**

β is the regression coefficient.

*P < 0.05.

**P < 0.01.

***P < 0.001.

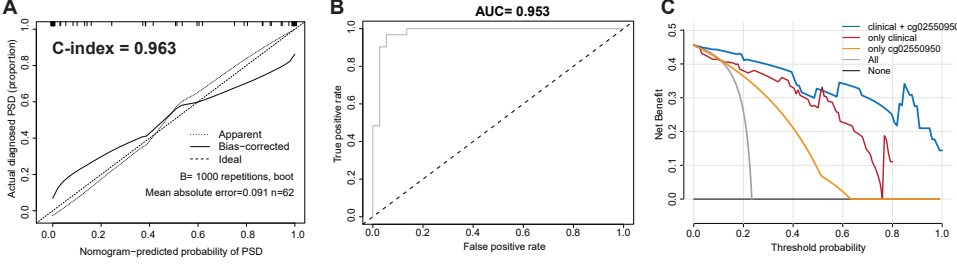

**Figure 2** **Validate the stability and accuracy of the PSD Nomogram in validation set.** (A) Calibration Diagram of PSD occurrence risk diagram model. (B) ROC diagram of the nomogram model for predicting the occurrence risk of PSD. (C) Clinical decision curve of PSD occurrence risk prediction model.

in Fig. 1E. the model is considered to have positive predictive abilities when the value of the AUC is between 0.5 and 1.0. The closer the ROC curve is to the upper left corner, the larger the AUC value, the higher the sensitivity, and the lower the misjudgment rate. The ROC value of the prediction model of the training dataset is as high as 0.913.

In order to verify the robustness and accuracy of the model, we repeated the above analysis in the validation set ($n = 62$). The calibration curve of the PSD nomogram demonstrated good agreement in the validation cohort (Fig. 2A). In addition, in the validation cohort, the C-index of the nomogram model and the AUC value of the ROC analysis were as high as 0.963 (95% CI [0.954–0.972]) and 0.953, respectively (Figs. 2A and 2B).

### Clinical practicability of the PSD risk prediction nomogram model

Taking into account the deviation of the model in clinical practice, this study uses a decision curve to evaluate the range of real benefits for patients, as shown in Figs. 1F and 2C. The decision curve shows that if this model is used when the threshold is 5–87%, the clinical benefit rate of the patients is the highest in both training and validation cohort. In addition, the combination of clinical features and methylation check points predicts a higher clinical benefit for PSD than either of them alone. Therefore, the prediction model developed in this study has high practical value in clinical settings.

## DISCUSSION

This was a prospective cohort study for the prediction of the incidence of PSD. This study incorporated a broader, more comprehensive, and novel predictor, combining the general condition of stroke patients, social-psychological factors, clinical data, and epigenetic factors to establish a more comprehensive nomogram model for risk prediction of PSD.

In many recent studies, gender, stroke location, and stroke severity (NIHSS score) have been confirmed to be associated with the occurrence of PSD (*Ilut et al., 2017*; *Mayman et al., 2021*). A total of 159 males and 67 females were included in this study. Univariate- and multivariate-logistic regression analysis showed that gender was an important factor affecting the incidence of PSD, but it was not an independent risk factor for PSD. Through the observation of the nomogram, it can be seen that women are more likely to develop PSD than men, which may be related to the personality, hormone secretion, lifestyle, and social influence of female patients (*Poynter et al., 2009*). The results of this study showed that patients who experienced an anterior circulation stroke had a higher risk of depression. We know that cerebral blood is mainly provided by the vertebrobasilar artery system and the internal carotid artery system. The frontal lobe, temporal lobe, parietal lobe, and basal ganglia are supplied by the internal carotid artery system. These are all located in the first three parts of the cerebral hemisphere, and blood flow to them is called "anterior circulation", while the blood flow of the five parts of the posterior part of the brain, including the brainstem and cerebellum, is provided by the vertebrobasilar artery system (called "posterior circulation") (*Menshawi, Mohr & Gutierrez, 2015*). Some studies have shown that the frontal and temporal lobes of the anterior circulation are significantly related to the occurrence of post-stroke depression (*Price & Duman, 2020*). The prefrontal lobe is generally considered to have a greater influence on human cognition and emotion, mainly relying on the medial prefrontal cortex to process related emotional information. Some studies have shown that it may also be involved in the occurrence and development of depression by affecting the fronto-occipital tract pathway (*Nelson et al., 2018*; *Howard et al., 2019*). The temporal lobe plays an important role in the regulation of negative emotions. Many clinical-imaging studies have shown that the activation of the temporal lobe in patients with depression is significantly higher than that in others during negative emotional self-regulation (*Maggioni et al., 2019*). A morphometry-based study found that patients with post-stroke depression had significantly lower gray-matter volumes in the hippocampus and anterior cingulate gyrus than those in the non-depression group (*Hong et al., 2020*). These studies' results are consistent with the results of our study.

In this study, univariate analysis showed that NIHSS, BI, and Rankin scores were significantly associated with the occurrence of PSD in patients with post-stroke depression and non-depression group ($P < 0.05$). Multivariate logistic regression analysis showed that NIHSS score was an independent risk factor for the occurrence of PSD, and the higher the score, the higher the probability of PSD. According to the results of previous studies, the more severe the symptoms of stroke patients, and the higher the dependence on daily life, the higher the incidence of PSD (*Omura et al., 2018*). Some patients with stroke often have neurological defects, which lead to sequelae such as loss of limb function or language,

which affect their work, life, and social interactions, resulting in negative social emotions and leading to the occurrence of PSD.

The time from onset to hospitalization refers to the interval between onset and hospitalization, which is a new clinical factor considered in this study compared with other, previous studies on risk factors for PSD. Combined with the characteristics of many inpatients and difficult admission in our hospital, the inclusion of this factor actually reflects the patients' ability to obtain social support on the side. Generally speaking, the longer the patients wait for hospitalization, the weaker their ability to obtain social support. Some studies have shown that social-support factors are closely related to the occurrence of PSD (*Lin et al., 2019*). At the same time, the clinical symptoms of patients with acute stroke may worsen with the increase of waiting time for hospitalization, which may also increase the risk of PSD. The results of this study show that the time from onset to hospitalization is a risk factor for the occurrence of PSD. According to the results of the nomogram, it can be found that the length of time is positively correlated with the occurrence of PSD.

Elevated lipid profiles, especially high levels of low-density lipoprotein cholesterol (LDL-C), have been associated with an increased risk of cerebrovascular events. Emerging evidence suggests that dyslipidemia might also be linked to post-stroke depression (PSD). It's theorized that imbalanced lipid levels may contribute to neural inflammation, oxidative stress, and impaired neurotransmitter metabolism, potentially exacerbating the pathogenesis of PSD (*Towfighi et al., 2017*). In our study, the risk of PSD within 3 months of poor lipid control was significantly increased. However, the precise mechanistic links between lipid control and the onset of post-stroke depression remain a topic of active investigation. Further large-scale, prospective studies are required to validate these associations and elucidate the underlying mechanisms.The results showed that the cg02550950 locus on the *HECW2* gene was significantly different between the PSD group and the non-PSD group. Multivariate logistic regression analysis showed that the methylation level of the cg02550950 locus on the *HECW2* gene was an independent risk factor for the occurrence of PSD. The nomogram of PSD risk prediction showed that the higher the degree of methylation of the cg02550950 locus on the *HECW2* gene, the greater the risk of PSD. Some studies have shown that the expression of Circ-*HECW2* can regulate miR-93 methylation, and then affect the growth and development (*Zuo et al., 2021*), and studies by *Krumm et al. (2015)* have shown that there is a link between mutations in the *HECW2* gene and the occurrence of autism. The studies of Liu X suggest that miR-93-5p may be involved in the occurrence of severe depression (*Liu et al., 2014*). Some studies have also confirmed that CircRNA*HECW2* inhibits the expression of miR-30d through the Notch/Notch1 pathway and promotes the production of *ATG5*, which aggravates endothelial interstitial transformation and leads to the destruction of the blood–brain barrier (*Yang et al., 2018*). This is also considered to be an important mechanism of bleeding conversion in patients with ischemic stroke, which aggravates the ischemic stroke condition (*Han et al., 2022*), and which may also indirectly lead to the occurrence of PSD. Although there is no direct, basic research to confirm the relationship between *HECW2* and PSD, combining this information with previous studies and our prospective study

undoubtedly provides a new direction for the study of the pathogenesis and therapeutic targets of PSD.

From the results of the nomogram, we can see that the risk factors related to the occurrence of PSD almost include the general situation of patients, psychosocial factors, and clinical factors. Not only that, but the difference in DNA-methylation level is also one of the important predictors, which indicates that the modeling results of this study are consistent with those of previous studies. Interestingly, the high methylation level of the cg02550950 site on the *HECW2* gene leads to the highest risk of PSD, indicating that although the pathogenesis of PSD is caused by both pathophysiological and social factors, pathophysiological factors may still be dominant. Finally, a variety of verification methods are used to evaluate the nomogram model for predicting the risk of PSD, including calibration curves, ROC curve analysis, C-index, and so on. The calibration curve is mainly used to evaluate the fitting of the predicted depression probability and the actual PSD-occurrence probability of stroke patients in the nomogram model. The internal repeated sampling technique (Bootstrap) is used to predict and analyze the queue-generating subset of the original data, and the C-index is as high as 0.908 (95% CI [0.901–0.915]), which shows that the model has good accuracy. The ROC curve analysis calculates the area under the ROC curve (AUC). The larger the area is, the stronger the ability of the model to accurately distinguish the occurrence of PSD. The AUC value of the model is 0.913, indicating that the prediction model has excellent distinguishing ability. To sum up, the nomogram model obtained in this study to predict the occurrence of PSD risk has a more accurate prediction ability.

This study established an early and comprehensive nomogram model for predicting the risk of PSD, making use of the clinical data and blood DNA-methylation levels, which are easily obtained from stroke patients in the early post-stroke stages, in order to help clinicians and patients recognize their own risk of PSD at the early stage of the disease, so as to carry out all aspects of early prevention and treatment. For example, patients can provide psychological comfort and music therapy to themselves, and doctors can use antidepressant prophylactic treatment in advance (*Zhang et al., 2013*). Nurses can also give patients more care and communication from the nursing level, providing tailored treatment to stroke patients with different conditions, so as to achieve the clinical purpose of reducing the risk of PSD and improving the prognosis of PSD patients.

## CONCLUSION

Our research not only provides a new predictive tool for the occurrence of PSD, but also provides a new starting point for the study of the pathogenesis of PSD. Although the PSD-prediction model based on patients' clinical characteristics and DNA methylation provides new therapeutic prospects for stroke patients, our study still has some limitations. Obtaining a large number of clinical samples, especially from different regions, remains a challenge. In addition, it should be noted that the causal mechanism of differential

methylation sites in PSD is still unknown, and further research is needed to study their functional role in the occurrence and development of PSD.

### Funding
This work was financially supported by the National Key Research & Development Program of China (grant number 2017YFC1310000). The funders had no role in study design, data collection and analysis, decision to publish, or preparation of the manuscript.

### Grant Disclosures
The following grant information was disclosed by the authors:
National Key Research & Development Program of China: 2017YFC1310000.

### Competing Interests
The authors declare there are no competing interests.

### Author Contributions
- Shihang Luo conceived and designed the experiments, performed the experiments, prepared figures and/or tables, authored or reviewed drafts of the article, and approved the final draft.
- Fan Liu conceived and designed the experiments, analyzed the data, authored or reviewed drafts of the article, and approved the final draft.
- Qiao Liao performed the experiments, authored or reviewed drafts of the article, and approved the final draft.
- Hengshu Chen performed the experiments, analyzed the data, prepared figures and/or tables, and approved the final draft.
- Tongtong Zhang performed the experiments, prepared figures and/or tables, and approved the final draft.
- Rui Mao conceived and designed the experiments, analyzed the data, prepared figures and/or tables, authored or reviewed drafts of the article, and approved the final draft.

### Human Ethics
The following information was supplied relating to ethical approvals (i.e., approving body and any reference numbers):

This study was approved by the Ethics Committee of Xiangya Hospital of Central South University (ethics approval number: 201910842).

### Data Availability
The code and raw data are available in the Supplementary Files.

### Supplemental Information
Supplemental information for this article can be found online at http://dx.doi.org/10.7717/peerj.16240#supplemental-information.

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
