# Peer review of "Nomogram model for predicting the risk of post-stroke depression based on clinical characteristics and DNA methylation"

_PeerJ, doi:10.7717/peerj.16240_

## Round 0.1 · original submission · Major Revisions

Authors should revise according to the suggestions of reviewer 1. The modifications should be marked. A point-to-point response letter is needed.

·

Basic reporting

The authors have carried out a prospective case-control study to model the occurrence of depression post-stroke. The reporting presented in the study needs to be improved to convey the essence of the study better.
(1) In one place the authors report that they used 10 cases + 10 controls to identify differentially methylated regions. In another place they mention that they used 226 recruits for the study. It is implied that the 10 cases + 10 controls were drawn from this sample, though this needs to be explicitly stated.
" 10 + 10 randomly selected samples for DNA methylation sequencing.
In this experiment, samples from 226
152 patients in the experimental and control groups were collected to verify and screen the differential
153 methylation sites"

(2) The identification of methylated regions may be better described. What is the effect size and p-values used? any thresholds were used? The authors state, "Based on their P-143 values, the site with the greatest difference in methylation degree and three sites with smaller P-144 values were selected." which is confusingly worded. All the information and the results of differential methylation analysis need to be provided.
(3) Some Results are presented in the Methods section.
(4) The authors have developed a risk-prediction model, but in essence it is a binary classification problem, and it may benefit the reader to note this simple formulation of the study.
(5) Typographical errors, namely formatting (of white-spaces), may be fixed. Esp. Table 2: Many values towards the end are missing (alignment problem?).
(6) Tables: The authors may explain Table 1 better. 300bp used -- but 30 kb mentioned in the text. Two additional sites shown in Suppl Table 1 -- what are these two sites? Table 3. Numerator for OR my be mentioned

Experimental design

There is an issue to consider with the experimental design of the study, resolving which is vital and necessary. The authors have determined methylated regions from a randomly selected subset of the cohort, which they then use to build the model. Since the methylated regions are indicative of case vs control (the question at hand -- PSD or non-PSD), there is significant data leakage in the model. It may not be surprising that the methylated region feature has a huge coefficient in the final nomogram model. The authors may address the reviewer's interpretation for the improvement of their experiment design.
Second, the criteria for identifying methylated regions appear weak and unsatisfactory.
Third, the authors may clarify the ambiguity regarding PSD diagnosis after one month or end of study after three months:
Depressive symptoms
102 were diagnosed with the fifth edition of the Diagnostic and Statistical Manual of Mental Disorders
103 (DSM-5) one month (±3 days) after stroke
In ABSTRACT:
Methods: We continuously recruited 226 patients
with a history of acute ischemic stroke and followed up for three months. Sociodemographic
indicators, vascular-risk factors, and clinical data were collected at
admission, and the outcome of depression was evaluated at the third month after stroke

Validity of the findings

The model validation is performed using internal bootstrap. At the minimum, for good machine learning practices, the test set even if internal must be independent of the train set used to build the prediction model.
Second, not all variables in multivariate model are significant. If five features are significant in multivariate model, only five would be appropriate for downstream nomogram construction. But the authors have reverted to seven variables : "Seven predictors screened by LASSO regression were used as independent variables for nomogram construction...

These issues coupled with data leakage mentioned in the 'Experiment Design' section are likely to inflate the true performance measures of the developed models, including AUC, C-index and confidence intervals. Thus attrative results notwithstanding, serious issues remain before the manuscript can be considered further.

Additional comments

(1) Some patients with stroke often have neurological defects, which lead to sequelae such as loss of
286 limb function or language, which affect their work, life, and social interactions, resulting in
287 negative social emotions and leading to the occurrence of PSD.
Also: In view of the fact that the clinical
62 diagnosis of PSD is often insufficient, and the onset frequently occurs one or more months after
63 the stroke, most medical environments also lack enthusiasm for PSD.
--> This leads to a request for substantively motivating the study question
(2) Although there is no direct, basic research to confirm the
338 relationship between HECW2 and PSD, combining this information with previous studies and our
339 prospective study undoubtedly provides a new direction for the study of the pathogenesis and
340 therapeutic targets of PSD.
--> Is it possible to examine the significance of methylation of HECW2 with other datasets?
(3) In Conclusion: prediction model based on patients’ clinical characteristics and DNA methylation provides new
373 therapeutic prospects for stroke patients, our study still has some limitations
--> the manuscript describes a model for PSD, but not any therapeutic prospects for stroke itself?
(4) The authors further state that,
"Through the observation of the nomogram, it can be seen that women are more
259 likely to develop PSD than men, which may be related to the personality, hormone secretion,
260 lifestyle, and social influence of female patients.
--> but this is based on a flawed nomogram construction
(5) On smoking:
However, interestingly, patients who quit smoking had a
314 higher risk of developing PSD than those who smoked a small amount every day; and patients who
315 never smoked had a higher risk than both those who quit, and those who smoked. A foreign study
316 of 6,146 subjects found a link between smoking and severe vitamin-D deficiency.[25] Previous
317 studies have shown that there is a strong correlation between low vitamin-D levels and depression
318 in patients with acute ischemic stroke;[26] therefore, smoking is a risk factor for PSD.
320 early smoking can help patients eliminate part of the psychological burden; with the progression
321 of the disease, smoking increases the risk of PSD

==> All these passages are self-contradictory

Reviewer 2 ·

Basic reporting

no comment

Experimental design

no comment

Validity of the findings

no comment

Additional comments

This paper is well written and is recommended to be accepted.

---

## Round 0.2 · Minor Revisions

Authors should revise according to the suggestions of reviewers. The modifications should be marked. A point-to-point response letter is needed.

·

Basic reporting

The authors have addressed the suggestions re. reporting, and as a result, the manuscript is much clearer.

Experimental design

The authors have addressed many of the queries and suggestions, except one core concern.
Previously I had flagged data leakage in the authors' models:
"Since the methylated regions are indicative of case vs control (the question at hand -- PSD or non-PSD), there is significant data leakage in the model. It may not be surprising that the methylated region feature has a huge coefficient in the final nomogram model. The authors may address the reviewer's interpretation for the improvement of their experiment design."
The authors in their response have concurred that the samples used for detecting DM regions are used in the train: test split *again* :
"We deeply appreciate the reviewer's astute observation regarding potential data leakage in our study design. To clarify, while 20 of the samples used for methylation sequencing were indeed included in the total cohort of 226 samples, we conducted a subsequent reexamination of all 226 samples to mitigate biases. When constructing our model, we meticulously segregated the training and validation sets and employed internal bootstrap validation to enhance the robustness of our results. Beyond DNA methylation, we integrated several clinical features that might influence PSD risk."
Unfortunately bootstrap does not solve the highlighted issue. Nor does separation of the train and test (or validation) sets.
The concern is not resolved (or, I find the writing ambiguous): some of the same samples (used for feature selection) may end up in the test set. When this happens, there is data leakage. This is common in machine learning models, and is a serious threat to the reliability of the models. Any model with data leakage is not following the scientific method. This needs to be addressed. The model may be re-constructed.

Validity of the findings

The authors seem to have revisited the analysis based on the previous comments, and have improved the validity of the findings clearly. Some commentary on the new models derived may be worthwhile. For e.g, the multivariate model has been re-derived to yield an increased set of predictors. It would be interesting if the authors could clarify what has changed in the new model, and if such changes may be expected if with more data or different analysis. It may be ascertained if the derived models are nominally stable with respect to the available data.

Additional comments

1. The authors mention in response to one of the comments (related to lines 174-180):
"Specifically, the formula used was: mean(PSD group) - mean(non-PSD group). From these results, we selected the two sites with the most pronounced differences in methylation. Furthermore, we identified three additional sites based on their P-values, with the smallest P-values indicating the most significant methylation disparities. Hence, five critical methylation sites were identified: cg18843803, cg03329597, cg13557709, cg02550950, and cg25290307"
It would be useful to the reader to state the exact effect sizes and significances that were used /obtained in the final analysis. [I could not find this information anywhere]
2. l326: The calculation according to the proposed nomogram is thus 237 points, predicting a PSD rate of 0.535%
I would like to know, what is this example intended to convey? Would this prediction of borderline rate of PSD occurrence mean anything?

---

## Round 0.3 · accepted · Accept

The authors have addressed the reviewers' concerns properly and revised the manuscript accordingly. The manuscript can be accepted for publication in its current form.